# Delayed healthcare seeking and associated factors for common childhood illnesses among caregivers with under-five children who visited Yem special woreda public health facilities, Southwest Ethiopia, 2023

**Gamechu Atomsa Hunde**[1]*, **Kalkidan Fikadu Zeben**[2], **Tigist Demeke**[1]

1 Faculty of Health Sciences, Jimma University Institute of Health, School of Nursing, Jimma, Ethiopia,
2 Department of Pediatric Nursing, School of Nursing College of Medicine and Health Sciences Wachemo University, Hosanna, Ethiopia

* gladgameato@gmail.com

## Abstract

### Background

For under-five children, receiving timely and appropriate medical attention is crucial in preventing serious and fatal complications. Unfortunately, evidence shows that parents of young children frequently delay seeking care, contributing to the death of many kids before they even get to a medical facility.

### Objectives

The study aimed to assess delay in healthcare seeking and associated factors for common childhood illnesses among caregivers with under-five children visiting Yem special woreda public health facilities, 2023.

### Methods

A facility-based cross-sectional study was conducted among 333 caregivers of under-five children diagnosed with common childhood illnesses visiting Yem special woreda public health facilities. Systematic random sampling was employed, and data was collected using an interviewer-administered questionnaire. Delay was characterized as a long time (typically >24 hours) between disease onset and beginning of treatment. Data was entered into Epi Data version 4.7 and exported to Statistical Package for the Social Sciences version 25.0. Binary logistic regression model was fitted. Bi-variable and multivariable logistic regression analyses were conducted to identify the factors that influence the delayed healthcare seeking. Adjusted odds ratios with a 95% confidence interval were used to determine the associations. Statistically significant variables were identified based on a p-value < 0.05.

### Results

A total of 326 caregivers participated in the study with a response rate of 98%. The proportion of delayed health care seeking was 74.5%. Caregivers "wait-and-see" approach is the

**Data Availability Statement:** The authors presented all the data in the main manuscript.

**Funding:** The author(s) received no specific funding for this work.

**Competing interests:** The authors have declared that no competing interests exist.

primary reason for the delay followed by traditional home treatments. Child ≥ 12 months (AOR = 1.99, 95% CI: 1.11–3.57), rural residence (AOR = 2.41, 95% CI: 1.35–4.28), no community health insurance (AOR = 1.91, 95% CI: 1.07–3.42), traditional treatment (AOR = 2.98, 95% CI: 1.46–6.10), and initial self-medication at home (AOR = 2.73, 95% CI: 1.32–5.63) and perceiving illness as mild (AOR = 2.64, 95% CI: 1.28–5.42) were associated with delayed healthcare seeking.

## Conclusion and recommendation

The study showed delay in health care seeking for common childhood illnesses among caregivers was high. Hence, reducing delays necessitates the implementation of public education campaigns, collaboration with local organizations, and the provision of counseling for caregivers regarding childhood illnesses.

## Introduction

Childhood illnesses encompass any sickness, disability, or abnormality that occurs during the developmental period from fetus to adolescence [1]. Common childhood illnesses are infectious diseases that causes significant morbidity and mortality in under five children worldwide, especially in low- and middle-income nations [2]. Malaria, pneumonia, and diarrhea are among the most common infectious diseases that exhibit a high susceptibility in young children and a prominent contributors to childhood mortality [3,4]. These illnesses are treated with artemisinin-based combination therapy (ACT), Amoxicillin tablets, oral rehydration salts (ORS), and zinc respectively. In addition, prevention of these diseases requires access to essential interventions like nutrition, vaccination, breastfeeding, hygiene, reducing air pollution, and affordable drugs [5–7].

The burden of illness and death from treatable diseases in children can be mitigated by timely medical care [8], but caregiver delays in seeking medical care can severely impact children's health and diminish the efficacy of treatments [9,10]. Delayed healthcare seeking is often defined as exceeding 24 hours from onset of the illness to initiation of the treatment [11]. These delays are common and involve challenges such as disease recognition, decision-making, and transportation to healthcare facilities [12]. Even with the available and accessible healthcare services, delays can lead to preventable illnesses, increased mortality, and economic burdens [13,14]. Such lateness can result in severe and potentially fatal complications, significantly contributing to under-five mortality rates. Addressing these delays is essential for reducing childhood mortality and improving health outcomes [15–17].

Despite global efforts to reduce under-five mortality through various health interventions, significant disparities persist, particularly in low- and middle-income countries [18,19]. Sub-Saharan Africa has the worst rate, registering 74 deaths per 1,000 live births, 15 times higher than Europe/North America and 19 times higher than Australia/New Zealand respectively [18].

Common childhood illnesses such as pneumonia, diarrhea, and malaria collectively accounted for nearly 30% of global under-five deaths in 2019, with Africa being responsible for a million deaths annually due to these conditions [3,20]. In 2023, in Ethiopia, approximately 22.5% of under five children suffered from these common childhood illnesses [21]. Organizations and governments around the world continue to work towards mitigating these

issues through improving healthcare accessibility, enhancing healthcare education for caregivers, and promoting early intervention and preventive care for children [22]. World Health Organization (WHO) and the United Nations Children's Fund (UNICEF) have collaboratively formulated the Integrated Management of Newborn and Childhood Illness (IMNCI) strategy [23].

Ethiopia, while making notable strides in child health through initiatives such as the Integrated Management of Newborn and Childhood Illness (IMNCI), still faces high child mortality rates, especially in underserved populations. One critical factor contributing to these high mortality rates is the delay in seeking healthcare for common childhood illnesses [24–26]. Ethiopia has set a target to reduce the child mortality rate to below 20 per 1,000 live births by the year 2035, striving to end all preventable child deaths through high-impact interventions, community empowerment, and improved health-seeking behaviors among families [24].

Studies conducted across different nations indicate that mothers and primary caregivers tend to delay seeking healthcare for their children's illnesses. In Cameroon, 88.1% of caregivers experienced a delay in seeking healthcare for common childhood illnesses [27], while in Nepal, this delay was reported as 62.7% [28]. In Bhubaneswar, the delay was found to be 15.77% [29].

In Ethiopia, significant proportions of delay for common childhood illnesses have been reported, with 86.3% in Jeldu district, Oromia region, 73.5% in Addis Ababa, and 73% in Aneded district, Northwest Ethiopia [30–32]. Factors such as lack of access to healthcare, insufficient awareness of when to seek medical help, knowledge of childhood illness, previous experience of childhood illness and economic barriers contribute to delays in seeking healthcare for children. Additionally, cultural or social factors in some regions may also play a role in delaying healthcare seeking [27,31,33–35].

Existing studies have documented the prevalence of healthcare-seeking delays in various regions of Ethiopia, highlighting the need for timely medical intervention to prevent severe and potentially fatal complications [33,34]. However, there is a lack of comprehensive understanding regarding the specific factors that contribute to these delays across different cultural, environmental, and socioeconomic contexts within the country [36]. Previous research has often focused on isolated regions or specific illnesses, leaving a gap in the broader understanding of how various determinants interact and influence healthcare-seeking behavior in diverse settings.

Moreover, there is limited evidence on the effectiveness of current interventions aimed at reducing delays in healthcare-seeking among caregivers of under-five children. Understanding these factors is crucial for designing targeted strategies that can effectively address the barriers to timely healthcare access and improve child health outcomes. Thus, this study aimed to assess delays in healthcare-seeking and associated factors for common childhood illnesses among caregivers of under-five children in Yem special woreda public health facilities.

## Methods and materials

### Study area and period

The study was conducted from June 1–30, 2023, in Yem Special Woreda, situated approximately 297 kilometers southwest of Addis Ababa. It comprises one town administration, three municipalities, and 34 rural kebeles. According to data from the Yem special woreda finance and development office in 2022, the total population of the area is 116,044 individuals, including 18,244 children under the age of five.

The Woreda has one primary hospital (Saja Primary Hospital) which serves a population of 32,234 and five health centers/ HC (Fofa, Semonama, Toba, Deri, and Gesi) [37]. The hospital operates with a total staff of 240 and offers a range of services, including one under-five

outpatient department (OPD), adult OPD, psychiatry OPD, emergency OPD, x-ray, ophthalmology OPD, ANC and family planning, delivery ward, neonatal intensive care unit (NICU), laboratory, ART and TB, pharmacy as well as adult and pediatric inpatient unit services. The HCs in the woreda provide primary healthcare services in under-five OPD, adult OPD, Maternity and delivery, laboratory, and pharmacy units.

## Study design

The study employed health facility-based cross-sectional study design.

## Study subjects and procedures

The study was conducted among caregivers of under-five children diagnosed with common childhood illnesses who visited under-five OPD at Yem special woreda public health facilities during the study period.

**Inclusion criteria.** Mothers/caregivers having under-five children who seek healthcare services at under-five OPD of Yem special woreda public health facilities and diagnosed with common childhood illnesses. For children diagnosed with more than one common illnesses, time-to-health seeking for the earliest illness to occur to child was recorded.

**Exclusion criteria.** Caregivers who came to the health facilities for appointments or follow up without new illness and whose child needed urgent referral were excluded from the study.

The largest sample size was considered after calculating for all the study objectives. Hence the final sample size was estimated using single proportion formula using the following assumptions:

$$n = \frac{(Z\alpha/2)^2 p(1-p)}{d^2}$$

Where,
n = the desired sample size
$Z\alpha/2$ = standard normal score at 95% (1.96)
p = 73% from a study of delay in seeking healthcare behavior for common childhood illnesses in Aneded district, Northwest Ethiopia [38].
d = margin of error (5%)

$$n = \frac{1.96^2 * 0.73(1-0.73)}{0.05^2}$$

By adding a 10% non-response rate, the final total sample size is **333**.

All public health facilities in the woreda are included in the study. The number of participants selected from each facility was proportionally allocated based on average monthly caseflow of last three months (March–May 2023) prior to data collection. Then a systematic random sampling technique was applied to select the eligible caregivers proportionally allocated to each health facility (*Fig 1*). The caregivers were interviewed keeping sampling interval of 2 between consecutive caregivers.

## Study variables

**Dependent Variable:** Delayed healthcare seeking.

**Independent variable. Socio-demographic factors:** caregivers' age, child age, child sex, monthly income, residence, family size, occupation, marital status, educational status, number of under-five children.

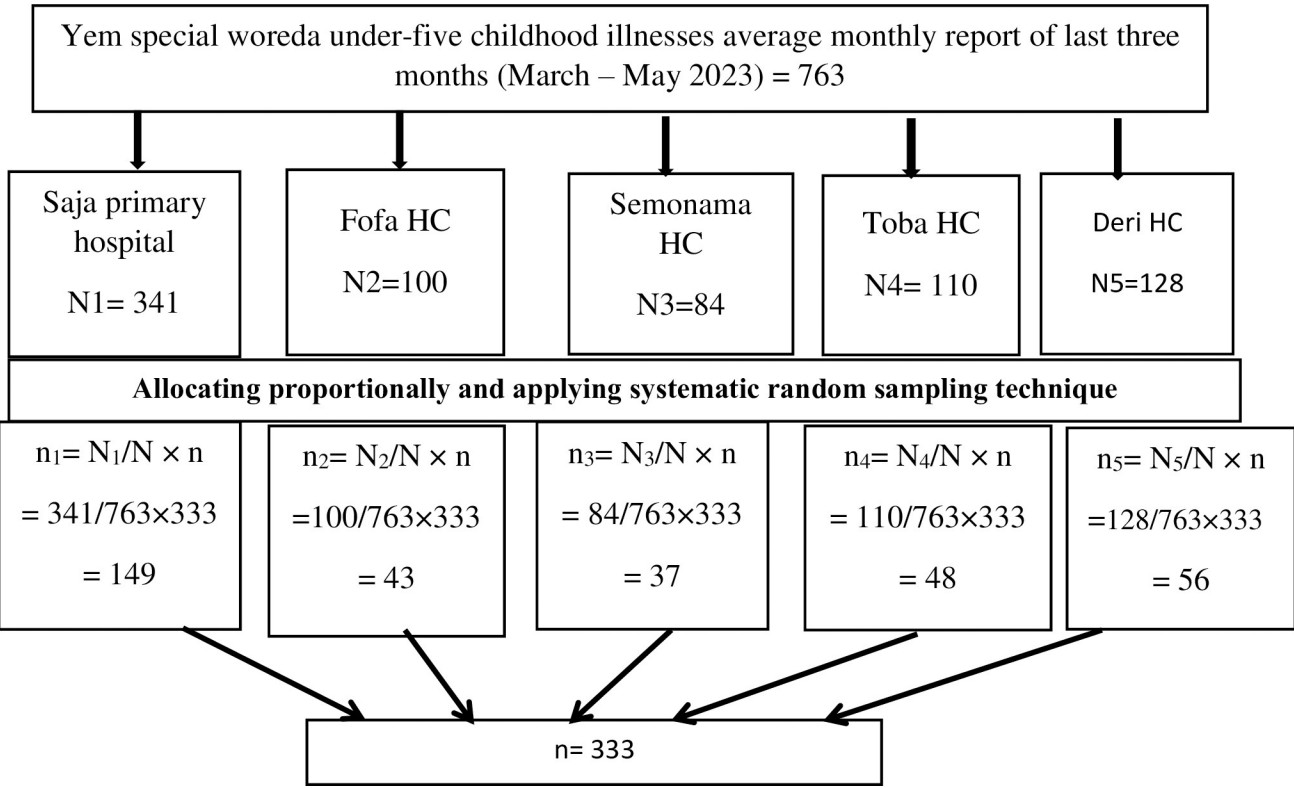

**Fig 1. Schematic presentation of the sampling procedure for assessing delayed healthcare seeking and associated factors among caregivers of under-five children visiting Yem special woreda public health facilities, Southwestern Ethiopia, 2023.**

**Caregiver's knowledge related:** knowledge of danger signs, perceived cause for recent illness, and perceived severity of illness.

**Child illness-related factors:** presenting symptoms, first choice of treatment for recent illness.

**Healthcare access-related factors:** type of health facility, distance to the nearest health facility, community health insurance.

**Caregivers' previous experience-related factors:** information about early seeking, a past illness history.

**Decision makers' behavior-related factors:** key decision maker, khat chewing status, smoking status, alcohol drinking status.

## Measurements

**Health care-seeking:** getting care from a qualified health care professional at government health centers or private hospitals/ clinics. The caregivers were asked the time they took to seek healthcare after they recognized the onset of the illness on their child [39].

**Perceived severity of illness:** measured by subjective assessment of the child's level of discomfort by the caregiver; classified as mild, moderate, or severe based on the caregivers' rating [28].

**Neonatal Danger Signs**: identified by the WHO such as not able to breastfeed/ unable to drink or eat, vomits everything, had convulsions/convulsing now, lethargic, unconscious, fast breathing, severe chest in drawing, fever, low body temperature [40].

**Knowledge of danger signs**: In our study, caregivers were asked to mention childhood danger signs that warrant immediate care-seeking from a health facility. It is measured based on

the spontaneous responses against a list of nine general child danger signs on the questionnaire as stipulated by WHO. The total number of correct spontaneous responses to 9 items (danger signs) with a minimum score of 0 and maximum of 9 was used to measure knowledge of mothers about danger signs [41].

**Monthly income:** Caregivers are asked to estimate their average income in one month from all sources.

## Operational definitions

**Healthcare seeking delayed:** if a care for a child is sought from health facilities more than 24 hours after becoming aware of symptoms [30,33,34].

**Healthcare seeking not delayed**: if a care for a child is sought from qualified healthcare providers earlier than 24 hours after onset of the illness [30,32,34].

**Common childhood illnesses:** ARI, diarrheal disorders, and febrile illnesses are the common childhood illnesses in this study [30].

**First of choice treatment:** defined as the setting where a caregiver makes their initial contact after the child becomes unwell (health institutions, traditional healers, self-medication, religious places) [42].

**Self-medication:** is when mothers become aware of their children's sickness, begin all available treatments, and apply them without a healthcare professional's prescription [39].

**Traditional medicine:** traditional healers, '*wogesha*', herbalists, and magicians (holy water) handle patients with apparent illnesses and sickness using experience-based knowledge and practice [39].

**Good Knowledge of general danger signs:** if mothers were able to mention at least three of the nine danger signs [40,41].

**Poor Knowledge of general danger signs**: if mothers mention less than three danger signs out of the nine danger signs [40,41].

**Monthly income:** classified according to the Ethiopian civil service proclamation into three categories: low income (<3000 ETB), medium income (3000–7500 ETB), and high income (>7500 ETB) [43].

## Data collection tool and procedures

The questionnaire was developed by reviewing relevant literature [16,27,32,33]. A structured interview administered questionnaire was employed to collect the data. This questionnaire has 6 components: socio-demographic characteristics, caregiver's knowledge and child's illness-related factors, health care access-related factors, caregivers' previous experience-related factors, decision makers' behavior-related factors, and promptness of care seeking for illness. The questionnaire was prepared in English and translated to local languages Amharic and Yemissa. Face-to-face exit interview was conducted in a separated room by one Nurse who has Bachelor of Science holder in each health facility and supervised by two master's holder nurses.

## Data processing, analysis and presentation

The data was checked for completeness, consistency, and errors before proceeding to data processing and analysis. Data was entered into Epidata version 4.7 and then exported to SPSS software version 25.0 for data cleaning and further analysis. Descriptive analysis was performed to determine proportions, standard deviations, and frequencies. A binary logistic regression model was fitted, and bi-variable and multivariable analysis was done. All assumptions of binary logistic regression were checked. Multicollinearity was checked between independent variables by the variance inflation factor (VIF = 1.030) and tolerance (>0.841). Variables with

p-values less than 0.25 in the bi-variable analysis were taken to multivariable analysis to control confounding [44]. Backward Likely Ratio (LR) method was used in multivariable analysis and model fitness was checked using the Hosmer–Lemeshow goodness of fit with p-value found to be 0.769. Statistical association between dependent and independent variables was determined at P-value <0.05. Variables that are statistically significant were identified as factors for delay in health care seeking. Finally, the result of the analyses was presented in texts, tables, and charts accordingly.

## Data quality control

The questionnaire was translated into the local languages of Amharic and Yemissa by language experts and reviewed by subject matter experts. Then, the Amharic and Yemissa versions were translated back to English, to verify the consistency. Content validity was checked by the subject matter experts. A pretest was carried out with 5% of the sample size (17 caregivers) in Gesi HC a week before the actual data collection time. A reliability of the tool was tested using Cronbach's alpha, giving a value of 0.804, showing internal consistency of the tool to measure the variables. Additionally, the order of the questionnaire modified, and ambiguous items were rephrased based on the findings of the pretest. A one-day orientation was provided for data collectors and supervisors on the purpose of the study, how to approach study subjects, and how to use the questionnaire. The collected data were reviewed and checked for completeness everyday by the supervisors and weekly by principal investigator.

## Ethical consideration

Ethical approval was obtained from Institutional Review Board (IRB) of Jimma University Institute of Health with the letter Ref. No: JUIH/IRB/393/23. A formal letter for permission and support was obtained from Jimma University School of Nursing to Yem special woreda health office. A letter of permission and support was written to each health facility from Yem special woreda health office. The study participants were informed of the purpose of the study, the procedures, and the potential risks of the study. Respondents were also informed that they can refuse or discontinue participation at any time they want. Participation was completely voluntary, and confidentiality of the information was guaranteed by secret code and kept anonymously. Then participants were asked if they agree or not agree to continue with the interview. Then verbal informed consent was taken from the study participants before starting the interview and their agreement/disagreement was recorded on the checkbox of consent form of each questionnaire which was approved by IRB. The hard copy of the collected data was kept in locked cabinet and the soft copy of the data were secured by a password on the computer.

## Results

### Sociodemographic characteristics of participants and children

In this study, out of 333 caregivers offered to the interview, 326 caregivers who had children under the age of five with common childhood illnesses completed the interview making a response rate of 98%. Among the participants, 186 (57.1%) were between the 25–34 age group, with a mean (SD) age of 31.03 (± 6.535) years. One hundred ninety-two (58.9%) of the children brought to the health facilities were 12 months and older, with a mean (SD) age of 19 (±13.704) months. The majority 309 (94.8) of the participants were married. More than half 171(52.5%) of the participants had attended primary school, and 155 (47.5%) identified as housewives by their occupation. In terms of income, 190 (58.3%) participants reported low monthly incomes (<3000 ETB) *(Table 1)*.

**Table 1. Socio-demographic characteristics of the study participants and their children in Yem special woreda public health facilities, Southwestern Ethiopia, 2023 (N = 326).**

| Variables | | Frequency(n) | Percent (%) |
|---|---|---|---|
| Caregivers' age | 18–24 | 42 | 12.9 |
| | 25–34 | 186 | 57.1 |
| | 35–44 | 80 | 24.5 |
| | ≥45 | 18 | 5.5 |
| Childs' age | <12 | 134 | 41.1 |
| | ≥12 | 192 | 58.9 |
| Sex of child | Male | 156 | 47.9 |
| | Female | 170 | 52.1 |
| Marital status | Single | 4 | 1.2 |
| | Married | 309 | 94.8 |
| | Divorced | 4 | 1.2 |
| | Widowed | 9 | 2.8 |
| Caregivers' residence | Urban | 93 | 28.5 |
| | Rural | 233 | 71.5 |
| Educational status | No formal education | 18 | 5.5 |
| | Primary education | 171 | 52.5 |
| | Secondary education | 71 | 21.8 |
| | College and above | 66 | 20.2 |
| Occupational status | Housewife | 155 | 47.5 |
| | Merchant | 27 | 8.3 |
| | Government employee | 72 | 22.1 |
| | Farmer | 56 | 17.2 |
| | Others [A] | 16 | 4.9 |
| Family monthly income | Low income(<3000 ETB) | 190 | 58.3 |
| | Medium income | 116 | 35.6 |
| | High income (>7500) | 20 | 6.1 |
| Number of family members | 1–3 | 75 | 23.0 |
| | 4–5 | 172 | 52.8 |
| | >5 | 79 | 24.2 |
| Number of under-five children | 1 | 273 | 83.7 |
| | 2 and above | 53 | 16.3 |

[A]: Student, daily laborer, driver, and mechanic.

## Caregivers' knowledge and child-illness-related factors

Among the total participants, the majority 278 (85.3%) of the participants had poor knowledge of the danger signs of common childhood illnesses. Regarding the children's presenting symptoms, 183 (56.3%) had present with cough. Nearly one-third 104 (31.9%) of the participants perceived their child's illness was caused by microorganisms. One hundred twenty-five (38.3%) of the study participants expressed a preference to seek initial medical care for their sick children at health institution. More than one-third of participants 118 (36.2%) rated their child's illness as moderate *(Table 2)*.

## Caregivers' previous experiences related factors

More than half, 184 (56.4%) participants got information about early treatment seeking and healthcare providers are the main source of information for majority of them. One hundred

**Table 2. Caregivers' knowledge and child-illness-related factors in Yem special woreda public health facilities, Southwestern Ethiopia, 2023 (N = 326).**

| Variables | | Frequency(n) | Percent (%) |
|---|---|---|---|
| Knowledge of danger sign | Poor | 278 | 85.3 |
| | Good | 48 | 14.7 |
| Child's presenting symptoms * | Cough | 183 | 56.3 |
| | Fever (body feels hotter than in a normal case) | 128 | 39.4 |
| | Vomiting | 53 | 16.3 |
| | Diarrhea (three or more loose or watery stools per day) | 26 | 8.0 |
| | Others [A] | 10 | 3.1 |
| Caregiver's perceived cause of recent illness | Evil eyes | 29 | 8.9 |
| | Poor hygiene and sanitation | 102 | 31.3 |
| | Microorganisms | 104 | 31.9 |
| | Cold | 56 | 17.2 |
| | I don't know | 18 | 5.5 |
| | Others [B] | 17 | 5.2 |
| Caregiver's first choice of treatment for recent illness | Take to health institution | 125 | 38.3 |
| | Take to traditional treatment | 88 | 27.0 |
| | Self-treatment at home | 84 | 25.8 |
| | Did nothing | 29 | 8.9 |
| Caregivers' perception of the severity of the illness. | Mild | 112 | 34.4 |
| | Moderate | 118 | 36.2 |
| | Severe | 96 | 29.4 |
| Perceived severity indicator (n = 96) | The child refused to suck my breast or eat foods | 38 | 39.6 |
| | The illness continues for a long time | 32 | 33.3 |
| | When a child's behavior changes | 22 | 22.9 |
| | Others [C] | 4 | 4.2 |

*: Variable with multiple responses

[A]: Difficulty of breathing, lethargy

[B]: Curse from God, teething, weaning

[C]: Fast breathing, blood stool, vomiting.

thirty (39.9%) participants reported that their children encountered illness in the last 6 months. Most of the caregivers, 90 (69.2%) visited a health facility for the illness their child encountered in the last 6 months. Of 130 participants who reported that their child previously got ill in the last 6 months, 40 (30.8%) of them didn't take their child to a health facility. Of these, 18 (45%), 11 (27.5%) of them managed their child by self-treatment at home and traditional medicine respectively *(Table 3)*.

## Healthcare access-related factors

In this study more than half 169 (51.8%) of the participants lived at more than 60 minutes of walking distance from a health facility, in contrast, 157 (48.2%) participants lived within a walking distance of 60 minutes or less from a health facility. The majority, 303 (92.9%) of the participants expressed their preference for seeking healthcare at government health facilities while 23 (7.1%) participants indicated a preference for private health facilities. Out of 23 participants who prefer private facilities, 18 (78.3%) of them believe treatment in the private is more effective and the remaining being nearby private facilities. Regarding community health insurance, 196 (60.1%) of participants had no community health insurance whereas 130 (39.9%) of participants had community health insurance.

**Table 3. Caregivers' previous experiences-related factors in Yem special woreda public health facilities, Southwestern Ethiopia, 2023 (N = 326).**

| | Variables | Frequency(n) | Percent (%) |
|---|---|---|---|
| Get information about early health care seeking | Yes | 184 | 56.4 |
| | No | 142 | 43.6 |
| Source of information [*] | Health care provider | 147 | 80.3 |
| | Radio | 20 | 10.9 |
| | Television | 9 | 4.9 |
| | Family | 10 | 5.5 |
| | Neighbor | 11 | 6.0 |
| | Social media (Facebook, Telegram, YouTube) | 6 | 3.3 |
| Child illness in the last 6 months | Yes | 130 | 39.9 |
| | No | 196 | 60.1 |
| Visited a health facility for the previous illness (n = 130) | Yes | 90 | 69.2 |
| | No | 40 | 30.8 |
| Previous visit help for today's visit(n = 90) | Yes | 65 | 72.2 |
| | No | 25 | 27.8 |
| How the previous visit helped (n = 65) | Counseled about the importance of visiting a health facility for childhood illnesses | 43 | 66.2 |
| | Told the danger of not visiting a health facility | 22 | 33.8 |
| If not visited health facility, how was the child treated of illness (n = 40) | Take to traditional treatment | 11 | 27.5 |
| | Self-treatment at home | 18 | 45.0 |
| | Treat the child with drug buying from pharmacies or drug sellers without Prescriptions | 7 | 17.5 |
| | Resolved by itself | 4 | 10.0 |

[*]: Variable with multiple responses.

## Decision makers' behavior-related factors

One hundred seventy-nine (54.9%) of the participants reported that mothers held the primary responsibility for making decisions regarding medical treatment for their children. Most 260 (79.8) of the decision makers do not consume khat and 265 (81.3) do not smoke cigarettes. Nearly half 155 (47.5) of the decision makers reported not consuming alcohol in the last 13 months (*Table 4*).

**Table 4. Decision makers' behavior-related factors in Yem special woreda public health facilities, Southwestern Ethiopia, 2023 (N = 326).**

| Variables | | Frequency(n) | Percent (%) |
|---|---|---|---|
| Decision maker | Mother | 179 | 54.9 |
| | Father | 82 | 25.2 |
| | Both mothers and father | 46 | 14.1 |
| | Grandparent | 19 | 5.8 |
| Khat chewing | Yes | 66 | 20.2 |
| | No | 260 | 79.8 |
| Smoking status | Every day | 34 | 10.4 |
| | Some days | 27 | 8.3 |
| | Not at all | 265 | 81.3 |
| Alcohol consumption in last 12 months | Almost every day | 59 | 18.1 |
| | At least once a week | 82 | 25.2 |
| | Less than once a week | 30 | 9.2 |
| | None in the last 13 months | 155 | 47.5 |

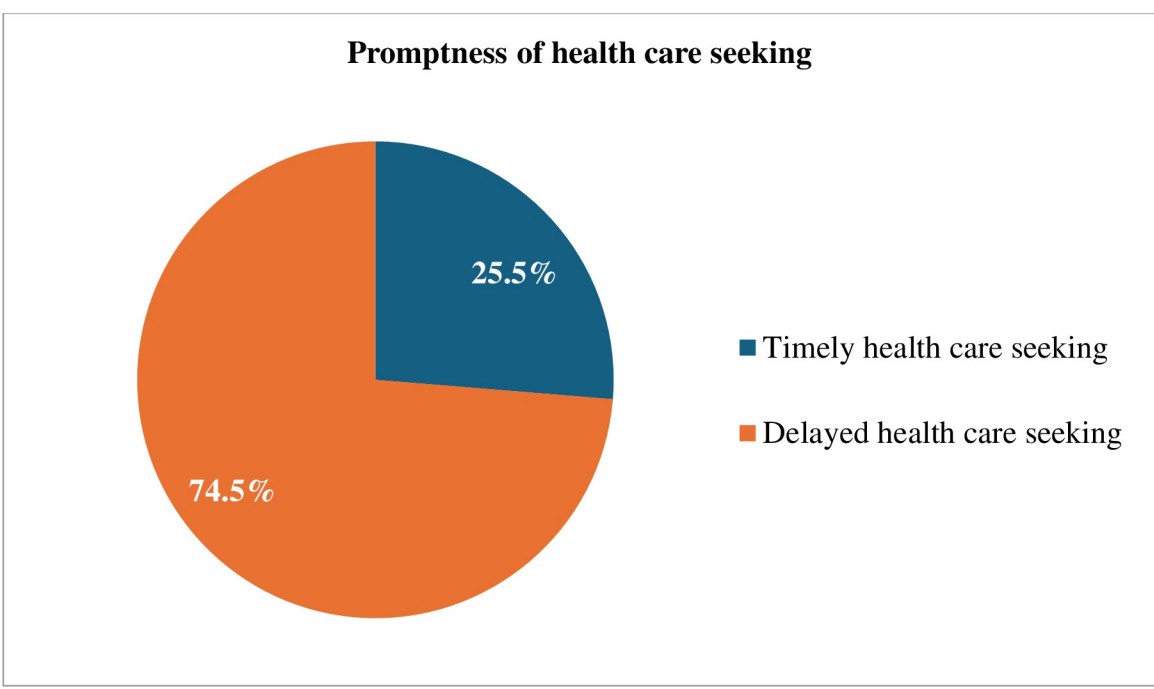

**Fig 2. Promptness of healthcare-seeking among caregivers in Yem special woreda public health facilities, Southwestern Ethiopia, 2023 (N = 326).**

### Promptness of healthcare-seeking

About 243 participants reported the signs /symptoms of the illness started more than 24 hrs before the time of seeking care. Therefore, the proportion of delayed healthcare seeking among the study participants was 74.5% (95% CI: 69.8–79.3). More than half 176 (54.0%) of caregivers arrived 2–3 days after symptoms started. The primary reason behind the delay in seeking healthcare was the expectation among 163 (67.1%) participants that the illness would improve by itself. Additionally, other significant reasons for the delay encompassed a reliance on traditional treatments 27 (11.1%), unable to decide by themselves 19 (7.8%), treatment with over-the-counter medication 6 (2.5%), and financial constraints experienced by 28 (11.5%) participants (*Fig 2*).

**Factors associated with delayed healthcare seeking for common childhood illnesses among caregivers with under-five children.** In this study, the odds of delayed health care seeking were two times (AOR = 1.99, 95% CI: 1.11–3.57) higher among caregivers who had a child ≥ 12 months than caregivers who had <12 months. The finding also revealed caregivers who reside in rural had about two and half times (AOR = 2.41, 95% CI: 1.35–4.28) higher odds of delayed health care seeking than those who reside in urban. On the other hand, the odds of delayed care-seeking were three times (AOR = 2.98, 95% CI: 1.46–6.10) higher for caregivers who used traditional treatment first, and 2.7 times (AOR = 2.73, 95% CI: 1.32–5.63) higher for those who tried self-medication at home first, as compared to going to a health facility first.

In similar ways the finding of this study revealed that caregivers who perceived child illness as mild had 2.6 times (AOR = 2.64, 95% CI: 1.28–5.42) higher odds of delayed health care seeking than caregivers who perceived it as severe. Furthermore, the odds of delayed health care seeking were approximately two times (AOR = 1.91, 95% CI: 1.07–3.42) higher among care givers who have no community health insurance than those who have community health insurance (*Table 5*).

**Table 5. Bi-variable and multi-variable logistic regression analysis of delay in health care seeking for childhood illnesses among caregivers of under-five children in Yem special woreda public health facilities, Southwestern Ethiopia, 2023 (N = 326).**

| Variables | | Delayed | healthcare seeking | COR (95% CI) | AOR (95% CI) |
|---|---|---|---|---|---|
| | | P- | value | | |
| YES | NO | | | | |
| Child age | ≥12months | 157 | 35 | 2.50(1.51–4.16) | **1.99(1.11–3.57)** | .022* |
| | <12months | 86 | 48 | 1 | 1 | |
| Caregiver residence | Rural | 187 | 46 | 2.69(1.59–4.54) | **2.41(1.35–4.28)** | .003* |
| | Urban | 56 | 37 | 1 | 1 | |
| Family monthly income | Low income (<3000 ETB) | 147 | 43 | 1.84(.69–4.90) | 1.77(.58–5.41) | .317 |
| | Medium income (3000–75000 ETB) | 83 | 33 | 1.35(.496–3.69) | 1.30(.43–3.96) | .641 |
| | High income (>7500) | 13 | 7 | 1 | 1 | |
| Knowledge of danger signs | Poor | 212 | 66 | 1.76(.92–3.38) | 1.47(.70–3.07) | .310 |
| | Good | 31 | 17 | 1 | | |
| First choice of treatment for recent illness | Traditional treatment | 74 | 14 | 3.19(1.62–6.26) | **2.98(1.46–6.10)** | .003* |
| | Self-treatment at home | 70 | 14 | 3.01(1.53–5.94) | **2.73(1.32–5.63)** | .007* |
| | Did nothing | 21 | 8 | 1.58(.65–3.86) | 1.01(.38–2.66) | .985 |
| | Health institution | 78 | 47 | 1 | 1 | |
| Information about early health care seeking | No | 111 | 31 | 1.41(.85–2.35) | 1.11(.62–1.99) | .715 |
| | Yes | 132 | 52 | 1 | 1 | |
| Caregiver's perceived severity of the illness | Mild | 96 | 16 | 3.44(1.76–6.75) | **2.64(1.28–5.42)** | .008* |
| | Moderate | 86 | 32 | 1.54(.86–2.76) | 1.31(.70–2.46) | .406 |
| | Sever | 61 | 35 | 1 | 1 | |
| Distance to the health facility (within walking distance) | >60 minutes | 131 | 38 | 1.39(.84–2.28) | 1.08(.60–1.94) | .809 |
| | ≤60 minutes | 112 | 45 | 1 | 1 | |
| Having community health insurance | No | 159 | 37 | 2.35(1.42–3.91) | **1.91(1.07–3.42)** | .028* |
| | Yes | 84 | 46 | 1 | 1 | |

1: Reference category, AOR: Adjusted odd ratio, COR: Crude odd ratio, CI: Confidence interval, Bold*-P value <0.05.

## Discussion

In this cross-sectional study, it was noticed that 74.5% (95% CI: 69.8–79.3) of caregivers delayed seeking healthcare for common childhood illnesses in children under the age of five. These findings indicate a substantial proportion of caregivers exhibit a delay in seeking necessary healthcare.

The current finding is consistent with previous studies conducted in the Aneded district of Northern Ethiopia, and Addis Ababa where the proportion of delayed healthcare seeking for

common childhood illnesses was reported at 73% and 73.3%, respectively [31,32]. This consistency may be attributed to caregivers' similar perceptions of illness severity within the Ethiopian context, where the perception of an illness as mild often leads to delayed healthcare seeking [33].

The results of this study indicate a higher proportion of delayed healthcare-seeking among caregivers compared to the study conducted in Nepal, which reported a proportion of 62.7% [28]. This disparity might be due to variations in caregivers' preferences for healthcare facilities as evidence shows caregivers utilizing public health facilities are more likely to delay seeking medical care compared to those utilizing private health facilities [36]. The findings of the present study indicated a significant preference among caregivers for public health facilities compared to the previous study. It could also be attributed to cultural differences in childcare and differences in socioeconomic status and health system.

Similarly the current study is higher than the study conducted in Bhubaneswar, where the proportion was 15.77% [29]. This difference could be due to variations in awareness creation efforts. In the present study area, the lack of regular organization of role plays and awareness camps was noted, unlike in the previous study area, where such initiatives might have contributed to lower delays in seeking healthcare [29].

In contrast, the present study revealed a lower proportion of delayed healthcare seeking when compared to previous studies conducted in the Jeldu district, Oromia region, where a proportion of 86.3% was reported [30]. The variation observed may be attributed to the difference in educational status among caregivers, as evidence shows more delayed healthcare-seeking among caregivers with lower educational levels [16]. The earlier study documented a notably higher percentage of caregivers with lower level of education in compared to the present study [30].

The proportion of delayed healthcare seeking in current study is also lower than a study conducted in Cameroon, which reported a proportion of 88.1% [27]. This disparity could be attributed to differences in the timing of the studies, as the previous research was conducted during the COVID-19 pandemic, which resulted in widespread disruptions to healthcare systems, including decreased access to healthcare facilities and concerns about contracting the virus [45,46].

The findings of this study revealed caregivers of children $\geq$ 12 months were more likely to delay seeking healthcare compared to caregivers of infants below 12 months. This observation aligns with studies conducted in Nekemte [33] and Tanzania [47]. The underlying reasons for this may be attributed to caregivers often prioritize the care of younger infants, holding the belief that children older than one year of age are more resilient to illness [33]. Furthermore, caregivers may possess an awareness that younger children are more susceptible to severe illnesses, contributing more prompt healthcare-seeking behavior for younger infants less than 12 months [48,49].

The present study revealed a higher likelihood of healthcare-seeking delay among caregivers residing in rural areas compared to those in urban settings, which aligns with findings from previous studies conducted in Nekemte town [33], Bahir Dar city [34], Hawassa [50], Kaduna Nigeria [35], and India [51]. This might be due to variations in the accessibility of public health services, resulting from geographical distance and media exposure [52]. Moreover, it could also be due to challenges posed by poverty and limited healthcare access in rural areas [53]. In contrast, urban households tend to benefit from better accessibility to healthcare services [54].

Additionally, the findings of this study showed that caregivers who lack community health insurance are more likely to delay seeking healthcare in comparison to those who have community health insurance. This result is in line with previous studies conducted in Nekemte

and Debre Markos town [33,55]. The reason might be being uninsured creates concerns about the financial burden associated with seeking healthcare as community-based health insurance significantly reduces healthcare costs [33,56].

Furthermore, the findings of this study showed that the odds of delayed healthcare-seeking are increased when caregivers resort to traditional medicine or self-medication. This result aligns with previous studies conducted in Bahir Dar and Nekemte [33,34]. The underlying reasons for this can be attributed to cultural beliefs that promote the use of home remedies and traditional medicine as the initial and acceptable form of treatment, as well as the community's adherence to religious beliefs and practices [36]. Moreover, caregivers' perceptions regarding the effectiveness of traditional treatments and the cost-effectiveness and accessibility of home remedies may also contribute to this trend [57,58].

Finally, consistent with a study conducted in Nekemte [33], the present study identified that caregivers who perceived their child's illness as mild exhibited a higher likelihood of delayed healthcare-seeking, compared to those who perceived the illness as severe. This might be due to caretakers tend to seek care from health facilities more frequently when they believe their children are seriously ill, as they want to prevent any potential complications [28]. Moreover, this phenomenon may be ascribed to caregivers' conviction that their child's illness will spontaneously ameliorate over time, prompting them to adopt a "wait-and-see" approach until further symptoms manifest [13].

Generally, a comprehensive strategy to enhance timely healthcare-seeking for childhood illnesses is paramount in reducing child mortality and morbidity. This includes implementing public education campaigns to raise awareness about the importance of early care, establishing partnerships with community leaders to promote timely healthcare as a social norm, and encouraging the renewal of health insurance for better access to services. Healthcare providers should prioritize counseling caregivers on illness management, the risks of relying on home remedies, and the importance of early treatment for children, including older ones. Additionally, further qualitative research is encouraged to explore cultural factors that contribute to delays in seeking care in different communities.

## Conclusion

The study revealed a high prevalence of delayed healthcare seeking among caregivers for common childhood illnesses. Key factors associated with these delays included having an older child (over 12 months), residing in rural areas, using traditional medicine or self-medication, lacking health insurance coverage, and perceiving the illness as mild. These findings suggest that these factors significantly hinder timely access to healthcare for children with common illnesses, highlighting areas for targeted interventions to improve healthcare-seeking behaviors and reduce delays.

## Limitations of the study

This study relies on self-reported data from caregivers, which might introduce recall bias. Details were not provided about specific traditional treatments or home remedies used.

## Acknowledgments

The authors are grateful to Jimma University, Institute of Health, Faculty of Health Sciences, School of Nursing. We also want to express our sincere appreciation to the Yem special woreda health office and the stakeholders in health facilities for their cooperation. We would like to thank the data collectors and supervisors. Finally, we are thankful to the study participants who generously volunteered their time.

## Author Contributions

**Conceptualization:** Gamechu Atomsa Hunde, Kalkidan Fikadu Zeben.

**Data curation:** Gamechu Atomsa Hunde, Kalkidan Fikadu Zeben.

**Formal analysis:** Tigist Demeke.

**Funding acquisition:** Kalkidan Fikadu Zeben.

**Investigation:** Kalkidan Fikadu Zeben.

**Methodology:** Gamechu Atomsa Hunde, Kalkidan Fikadu Zeben, Tigist Demeke.

**Project administration:** Kalkidan Fikadu Zeben.

**Supervision:** Kalkidan Fikadu Zeben, Tigist Demeke.

**Validation:** Kalkidan Fikadu Zeben, Tigist Demeke.

**Writing – original draft:** Gamechu Atomsa Hunde, Kalkidan Fikadu Zeben.

**Writing – review & editing:** Gamechu Atomsa Hunde, Tigist Demeke.

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
