## [Decision Letter · Decision Letter 0]

22 Mar 2024

PONE-D-24-00553Delayed Healthcare Seeking and Associated Factors for Common Childhood Illnesses among Caregivers with Under-Five Children in Southwestern Ethiopia, 2023.PLOS ONE

Dear Dr. HUNDE,

Thank you for submitting your manuscript to PLOS ONE. After careful consideration, we feel that it has merit but does not fully meet PLOS ONE’s publication criteria as it currently stands. Therefore, we invite you to submit a revised version of the manuscript that addresses the points raised during the review process.

We look forward to receiving your revised manuscript.

Kind regards,

Dereje Oljira Donacho, PhD in Environmental Health

Academic Editor

PLOS ONE

2. In the ethics statement in the Methods, you have specified that verbal consent was obtained. Please provide additional details regarding how this consent was documented and witnessed, and state whether this was approved by the IRB.

Reviewers' comments:

Reviewer's Responses to Questions

**Comments to the Author**

1. Is the manuscript technically sound, and do the data support the conclusions?

Reviewer #1: Yes

Reviewer #2: Partly

2. Has the statistical analysis been performed appropriately and rigorously? 

Reviewer #1: No

Reviewer #2: Yes

3. Have the authors made all data underlying the findings in their manuscript fully available?

Reviewer #1: No

Reviewer #2: No

4. Is the manuscript presented in an intelligible fashion and written in standard English?

Reviewer #1: Yes

Reviewer #2: Yes

5. Review Comments to the Author

Reviewer #1: Comments and suggestions from a reviewer on article entitled with>

Delayed Healthcare Seeking and Associated Factors for Common Childhood illnesses among Caregivers with Under-Five Children in Southwestern Ethiopia, 2023.

General Comment

>INITIALLY I am happy to evaluate such important paper that address the real problem of our community, BUT the paper have a lot of minor and major problems on result and discussion part……by saying this I have the following concerns,

1. Please clearly write your operational definition

2. How you assess Caregivers’ perception of the severity of the illness. Its not well defined

3. line 274 please check the table inconsistence

4. Visited a health facility

(n=130)Yes 90 69.2 No 40 30.8 Previous visit help for today's visit

Its not clear for me because your study was institution based so ….

5. on Decision maker figure I have a doubt with the context of our country…

6. please check it the figure is exaggerated on substance like alcohol and smoking ….

7. Major issue is about analysis totally it needs revision

For example variable \\Distance to the

health facility

(within walking/ is not associated in binary…COR 1.39(.84-2.28), Information about

early health care

seeking, Knowledge of

danger signs………………….

8. your discussion is shallow…please elaborate more

Reviewer #2: I would like to extend my sincere appreciation to authors for presenting us this interesting paper, which aimed to determine the magnitude and identify factors associated with delayed healthcare seeking in southwestern part of Ethiopia. Timely treatment is crucial to get a favorable outcome, which intern help to reduce morbidity and mortality. The benefit of timely treatment is not only for positive prognosis of disease but it has an advantage cost-wise too.

General comments

Do you think your drawn samples are representative for the population made generalization (southwest Ethiopia)? In another word, can you confidently generalize your study finding to southwest Ethiopia, regardless of the difference in access to health facility, socio-economic status, and government administrative (zonal town, woreda town, ….) across the region?

How do you think your design is address your objective using a quantitative method only?

Do you think doing a cross sectional study using 333 samples is adequate?

The statistical analysis applied is not clearly elaborated under ‘Data processing, analysis and presentation’

Specific comments

Introduction

This study does not sound novel to me. Since there are so many studies on related topics worldwide, including Ethiopia. Therefore, the authors need to try harder to clarify the motivation and contributions of this study.

The introduction needs to be well organized. Similar paragraphs should be synthesized.

Why does southwest Ethiopia matter in this study?

The scientific background and rationale for the investigation is not clearly addressed; which you raised as a gap should be clearly stated.

Your introduction is not comprehensive; it did not covered all variables assessed in your study example knowledge of caregivers, behavior, previous experiences,…

I’m not clear with your objective. Is it to assess delays in healthcare-seeking and associated factors for common childhood illnesses among caregivers of under-five children in Southwestern Ethiopia or Yem special woreda public health facilities?

Method and materials

In the whole document, the authors stated the study setting as southwest by conducting in only one special woreda, even though there are many health facilities in southwest Ethiopia. Do you think it is scientifically plausible? How you select the current study setting?

Unpublished data should be acknowledged.

Inclusion and exclusion criteria should be well described. For instance, what you were done when the child came with two or more diagnosis?

You missed one important sub-heading, ‘Study Design’. it should be an independent sub-heading and clearly stated.

Can you justify why you applied systematic sampling method? And did you believe no other appropriate method than SS?

You missed one important sub-heading, ‘Measurements’ and mixed it with Operational definition.

Page 7, Is the data collection tool standardized? How?

Page 8, line 208-9, Model fitness was checked using the Hosmer–Lemeshow goodness of the model with p-value found to be greater than 0.05.” instead of this you should put the actual result?

What is your method of variable selection for your multivariable analysis?

How have you tried to control possible confounding and biases?

Page 8, line 209-11, rephrase the sentence and remove the redundant statement.

Page 8, line 217-19, authors mentioned pretest and validity test was carried out. Could you supplement them with results? What modification or revision did you do for your tool?

Page 8, line 223, “The collected data were reviewed and checked for completeness everyday by the supervisors and principal investigator.” Could you elaborate who is principal investigator and who are supervisors?

A major revision is needed in the ‘Data processing and analysis’ sub-heading. The most critical information was missed.

Page 9, line 230, Rephrase this sentence “Verbal informed consent was taken from the study participants before starting the interview after were informed of the purpose, and benefits of the study.”

Page 9, “Ethical consideration” most of the sentences convey almost similar message. Could you rephrase them?

Results

Did the authors used the standard variable categorisation? Especially for ages, number of family,...

Page 13, Caregivers’ previous experiences-related factors out of 130 child visited a health facility only 40 of them cured of illness?

I believe that some findings are out of scope. If it is necessary, it should be included on the background and objective of the study.

If assessing substance use is necessary, it should be holistic. For example, you should include frequency, duration, amount,…

Approximate or round the numbers to the nearest while interpreting odds ratio. For instance, instead of saying 1.99 times, say 2 times….

Discussion, conclusions, and recommendations

Authors’ discussion is weak. Majority of the findings are not well justified.

Justification under the discussion at are not supported by evidence

There is no strength of the study stated.

No limitations of the study provided.

Authors missed important heading “conclusion”.

The author should recommend different stakeholders specifically based on the finding of the study, to alleviate the problem (high prevalence of delayed healthcare seeking).

References

Some of your references seem incomplete.

6. PLOS authors have the option to publish the peer review history of their article (what does this mean?). If published, this will include your full peer review and any attached files.

Reviewer #1: No

Reviewer #2: No

---

## [Author Response · Author response to Decision Letter 0]

28 May 2024

Response to reviewers is submitted separately as doc

---

## [Decision Letter · Decision Letter 1]

21 Aug 2024

PONE-D-24-00553R1Delayed Healthcare Seeking and Associated Factors for Common Childhood Illnesses among Caregivers with Under-Five children who visited Yem special woreda public health facilities, Southwest Ethiopia, 2023.PLOS ONE

Dear Dr. Hunde,

Thank you for submitting your manuscript to PLOS ONE. After careful consideration, we feel that it has merit but does not fully meet PLOS ONE’s publication criteria as it currently stands. Therefore, we invite you to submit a revised version of the manuscript that addresses the points raised during the review process. Please submit your revised manuscript by Oct 05 2024 11:59PM. If you will need more time than this to complete your revisions, please reply to this message or contact the journal office at plosone@plos.org. Please include the following items when submitting your revised manuscript:A rebuttal letter that responds to each point raised by the academic editor and reviewer(s). You should upload this letter as a separate file labeled 'Response to Reviewers'.A marked-up copy of your manuscript that highlights changes made to the original version. You should upload this as a separate file labeled 'Revised Manuscript with Track Changes'.An unmarked version of your revised paper without tracked changes. You should upload this as a separate file labeled 'Manuscript'.If applicable, we recommend that you deposit your laboratory protocols in protocols.io to enhance the reproducibility of your results. Protocols.io assigns your protocol its own identifier (DOI) so that it can be cited independently in the future. For instructions see: https://journals.plos.org/plosone/s/submission-guidelines#loc-laboratory-protocols. Additionally, PLOS ONE offers an option for publishing peer-reviewed Lab Protocol articles, which describe protocols hosted on protocols.io. Read more information on sharing protocols at https://plos.org/protocols?utm_medium=editorial-email&utm_source=authorletters&utm_campaign=protocols.

We look forward to receiving your revised manuscript.

Kind regards,

Dereje Oljira Donacho, PhD

Academic Editor

PLOS ONE

Journal Requirements:

Reviewers' comments:

Reviewer's Responses to Questions

**Comments to the Author**

1. If the authors have adequately addressed your comments raised in a previous round of review and you feel that this manuscript is now acceptable for publication, you may indicate that here to bypass the “Comments to the Author” section, enter your conflict of interest statement in the “Confidential to Editor” section, and submit your "Accept" recommendation.

Reviewer #3: (No Response)

Reviewer #4: (No Response)

2. Is the manuscript technically sound, and do the data support the conclusions?

Reviewer #3: Yes

Reviewer #4: Yes

3. Has the statistical analysis been performed appropriately and rigorously? 

Reviewer #3: Yes

Reviewer #4: Yes

4. Have the authors made all data underlying the findings in their manuscript fully available?

Reviewer #3: Yes

Reviewer #4: Yes

5. Is the manuscript presented in an intelligible fashion and written in standard English?

Reviewer #3: Yes

Reviewer #4: Yes

6. Review Comments to the Author

Reviewer #3: Introduction

Line 72- 81 the paragraph is not clear please try to rewrite with good and undrstandable grammer.

Method

The sampling procedure,the sampling method and the sample size calculation is not well writened

And as i see u have used multistage sampling technique so you need to have put how much the design effect

Better to put diagram to show the sampling procedure

Conclusion and recommendation

Your recommendation should inline with the conclusion better to re write

Reviewer #4: Manuscript Number: PONE-D-24-00553 Title: Delayed Healthcare Seeking and Associated Factors for Common Childhood Illnesses among Caregivers with Under-Five Children in Southwestern Ethiopia, 2023.

General comment:

The study was done on an important public health issue as Ethiopia is still in a plight of pervasive under-five child mortality. Thus, the manuscript can be considered for publication as it stands now.

However, I need some clarification, and if it is possible, with a correction suggestion on some idea or sentence in the manuscript that makes me confused and is not clear.

Specific comments

1. You were reporting both good and bad knowledge of danger signs. What is your base, to say, good or poor? Knowledge of danger signs for a particular study participant or caregiver means that how do you measure their knowledge and what is your evidence?

2. Delayed healthcare seeking is categorized as “yes” or "no." What is your base to categorize as “yes” or “no” because you are not clearly operationalized at the section of operational definition, just you are only giving a definition of what is "delay in healthcare-seeking" means?”

3. I try to see in detail your methodology section, so I feel that there is a missed sub-section, which is “study variable.” Under this section, you must clearly state your independent (explanatory) variable and dependent (outcome) variable in this section. Off course I understand that you are trying to put some independent and dependent variables under the Operational Definitions section, but your study variable must be put separately under the sub-section of the study variable. Then you can operationalize these variables, which makes them confused when someone reads your paper with their appropriate definition and how you measured them.

4. What are your inclusion and exclusion criteria? You missed this one also under methodology its good putting in separate subsection of inclusion and exclusion criteria.

5. Please add a citation or reference for your line 207 whey 0.25; why not 0.2? Some studies also use 0.2 as a cutoff point. Wherever, put your evidence for your cut point if the study recommends or uses a cut point of 25%.

6. Data quality control: What about Cronbach's alpha value to approve wither your data consistent or not?

7. Data processing, analysis: What about multicollinearity and correlation of your data to approve wither your data normal or not? And

8. Finally, I suggest carefully looking at the grammar error and all text formatting, like spacing and indentation, removing unnecessary spacing between paragraphs, and also checking the PLOS one manuscript submission format. I detect such a problem in your document.

In general, this paper is written well. Information and knowledge reflected in this study is valuable for policy development for countries and organizations and stakeholder groups who are working on child care, and it is also a good input for developing an inclusive starting plan for family health.

7. PLOS authors have the option to publish the peer review history of their article (what does this mean?). If published, this will include your full peer review and any attached files.

Reviewer #3: No

Reviewer #4: **Yes: **Fikadu Wake Butta

---

## [Author Response · Author response to Decision Letter 1]

28 Sep 2024

The authors are very grateful to the editor and reviewers for giving their precious time to critically review the manuscript and providing vital comments and suggestions to make our document more insightful. 

Below is the response to the concerns raised by the reviewers

Letter to editors

1. Borderlines removed from table 1, 2, 3, 4 and 5

2. References are edited.

Response to Reviewer #3

1. Line 72-81 paragraph is edited and rewritten. See Line 72-82

2. Population, sample size estimation and Sampling procedure edited. Diagrammatic presentation of sampling procedure inserted. See line150-174.

3. The study employed systematic random sampling technique. Since all the public health facilities in the Yem special woreda (the study area) were included in the study, Multistage sampling technique was not applied in the study. Therefore, the authors believe there is no variation in estimation requiring design effect.

4. Recommendation: Recommendation section is removed as per the journals guidelines and merged into discussion section. The suggested recommendations are edited and aligned with findings of the study. See line 449-458

Response to reviewer #4

1. Knowledge of danger signs was measured based on caregivers’ ability to spontaneously mention WHO danger signs. There is no universally agreed up on base to classify caregivers’ knowledge of danger signs. However, researchers use different cut-off points to classify knowledge. Based on this the authors classify knowledge based on the number of danger signs they are able to mention. Therefore, those caregivers who listed less than three (3) danger signs were classified to have poor knowledge of danger signs, while those who listed three and above danger signs are classified as having good knowledge of danger signs.

2. Reviewer’s comment: “Delayed healthcare seeking is categorized as “yes” or "no." What is your base to categorize as “yes” or “no” because you are not clearly operationalized at the section of operational definition, just you are only giving a definition of what is "delay in healthcare-seeking" means?”

Author’s response: Operational definition for delayed healthcare seeking and healthcare seeking not delayed separately defined in the revised manuscript. See line 211-214

3. Study variables subsection is separately inserted. See line 175-189

4. Eligibility criteria inserted as a separate subsection. See line 154-159

5. Comment addressed as the reviewer suggested. Reference inserted. See line 244-245

The use of a higher p-value threshold (e.g., 0.25) for variable selection reflects the idea that the bivariate analysis alone does not account for the effects of other variables. A less strict threshold helps to avoid discarding variables that could still have a meaningful impact when adjusted for other predictors in the multivariable model. Different statistical books suggest p-value as high as 0.25 for variable selection. Few are mentioned below:

In the step where variables are being screened for possible inclusion in the multivariable model, we recommend screening variables using a p-value of 0.25 for inclusion in the model.”

 (Hosmer & Lemeshow, 2000, p. 95).

“We suggest selecting variables with p-values of 0.25 or less in the bivariate analysis for inclusion in the multivariable model.”

(Bursac et al., 2008). Reference: Bursac, Z., Gauss, C. H., Williams, D. K., & Hosmer, D. W. (2008). Purposeful selection of variables in logistic regression. Source Code for Biology and Medicine, 3(1), 17. doi:10.1186/1751-0473-3-17

6. Reviewers comment: Data quality control: What about Cronbach's alpha value to approve wither your data consistent or not?

Authors response: The value of the Cronbach’s alpha was inserted. It is within acceptable range showing the internal consistency of the tool to measure the variables. see line 262

7. Reviewers comment: “Data processing, analysis: What about multicollinearity and correlation of your data to approve wither your data normal or not? And”

Authors response: Multicollinearity checked, and value inserted in the main manuscript. See line 243-246

8. Editorial checking: Grammar, editorial and punctuations are checked and corrected to the best of the author’s ability.

---

## [Editor Report · Decision Letter 2]

2 Oct 2024

Delayed Healthcare Seeking and Associated Factors for Common Childhood Illnesses among Caregivers with Under-Five children who visited Yem special woreda public health facilities, Southwest Ethiopia, 2023.

PONE-D-24-00553R2

Dear Dr. Hunde,

We’re pleased to inform you that your manuscript has been judged scientifically suitable for publication and will be formally accepted for publication once it meets all outstanding technical requirements.

Kind regards,

Dereje Oljira Donacho, PhD

Academic Editor

PLOS ONE
---

## [Editor Report · Acceptance letter]

8 Oct 2024

PONE-D-24-00553R2 

PLOS ONE

Dear Dr. Hunde, 

I'm pleased to inform you that your manuscript has been deemed suitable for publication in PLOS ONE. Congratulations! Your manuscript is now being handed over to our production team.

Kind regards, 

on behalf of

Dr. Dereje Oljira Donacho 

Academic Editor

PLOS ONE